# The Growth and Spectroscopic Properties of Er, Nd: YSGG Single Crystal Fibers

**Baiyi Wu** [1,2], **Tao Wang** [3], **Meng Wang** [1,2], **Jian Zhang** [3], **Ning Jia** [4], **Zhitai Jia** [3,*] and **Zefeng Wang** [1,2,*]

1    College of Advanced Interdisciplinary Studies, National University of Defense Technology, Changsha 410073, China; wubaiyi@nudt.edu.cn (B.W.)
2    Nanhu Laser Laboratory, National University of Defense Technology, Changsha 410073, China
3    State Key Laboratory of Crystal Materials, Shandong University, Jinan 250100, China
4    Key Laboratory of Materials for High Power Laser, Shanghai Institute of Optics and Fine Mechanics, Chinese Academy of Sciences, Shanghai 201800, China
*    Correspondence: z.jia@sdu.edu.cn (Z.J.); zefengwang@ndut.edu.cn (Z.W.)

**Abstract:** Single crystal fiber (SCF) is a novel solid gain medium and technique which combines the advantages of glass fiber and single crystal, showing great potential in the field of high-power lasers. In this paper, Er, Nd: YSGG single crystals with diameters of 2 mm and lengths of 80 mm were successfully grown using the micro-pulling-down method for the first time. Then, the measurements of Laue spots and $Er^{3+}$ distribution indicated that the as-grown crystals were of a high quality. The effect of co-doped $Nd^{3+}$ on the Er: YSGG was systematically discussed, which demonstrated that $Nd^{3+}$ can decrease the fluorescence lifetime of Er: $^4I_{13/2}$ that solve the self-termination bottleneck accordingly. These results demonstrate that Er, Nd: YSGG SCFs are promising materials for the further 3 µm laser generations.

**Keywords:** micro-pulling-down (µ-PD); growth of $Y_3Sc_2Ga_3O_{12}$ (YSGG); Er-doped crystal

## 1. Introduction

$Er^{3+}$-doped crystals are an important solid gain medium to obtain lasers operating around 2.7~3.0 µm because of the operation on the $Er^{3+}$: $^4I_{11/2} \rightarrow {}^4I_{13/2}$ transition. The growth of high-quality crystals and the optimization of laser output are attracting much attention to develop the 3 µm laser output [1–3]. Currently, the $Er^{3+}$-doped crystal laser is limited by a low absorption efficiency and self-terminating behavior during operation; moreover, high $Er^{3+}$ concentration doping will usually lead to thermal effects [4,5].

Single crystal fiber (SCF) is a novel gain medium material which combines a high aspect ratio and a high gain coefficient to achieve the desired laser output. Compared with traditional bulk crystals, the thermal management of the SCF laser system is better because of its high aspect ratio [6–10]. Garnet crystal fiber like $Y_3Al_5O_{12}$ (YAG) possesses high thermal conductivity, high doping concentration and a laser damage threshold which can obtain an over 50 times higher laser output than that of traditional glass fiber, according to the theoretical calculation [11]. An output power as high as 251 W with a slope efficiency of 53% has been achieved with YAG SCF, which is the record for CW (continuous wave) laser output of SCFs so far [12]. More importantly, heavily-doped and high-quality single crystal fibers can be prepared easily. There are two main methods to achieve such single crystal fibers; the features of the micro-pulling-down method (µ-PD) include the thermal field around the growing interface that can be controlled by the designed and optimal crucible, the after-heater and the thermal insulations. So, it can be used to obtain high-melting materials, novel materials and volatile materials [11,12]. The laser-heated pedestal growth method (LHPG) utilizes a focused $CO_2$ laser beam to heat the source rods to form a melting zone; the fast crystal growth can be achieved when the oriented seed crystal touches the

hemispherical melting zone. It can be used to obtain extremely high-melting materials and ultra fine crystal fibers [13,14].

Er: YSGG ($Y_3Sc_2Ga_3O_{12}$) is also a potential garnet crystal for obtaining 2.7~3 μm lasers due to its low phonon energy, excellent thermal conductivity and high laser damage threshold [15]. YSGG is a typical cubic crystal and belongs to Ia3d; and Er, Nd:YSGG has similar lattice constant to pure YSGG because the ionic radius of $Er^{3+}$ is close to that in $Y^{3+}$. The melting point of YSGG is 1877 °C and its thermal expansion coefficient is $8.1 \times 10^{-6}/°C$. Sc partially replaces Ga in YSGG, which not only reduces the defects caused by the segregation of $Ga_2O_3$ in the growth, but also makes it easier to achieve flat interface growth. It makes it easier to prepare high-quality and high-doped Er: YSGG single crystal fiber. Introducing the other rare-earth ions is an efficient method to solve low absorption efficiency and self-terminating behavior during operation. For example, sensitized ions like $Yb^{3+}$ and $Cr^{3+}$ can be doped to enhance the absorption of the pump energy at 940 nm, and the absorption of $Cr^{3+}$ at 450 nm and 634 nm, corresponding to the energy level transitions of $Cr^{3+}:^4A_2 \rightarrow {}^4T_1$ and $Cr^{3+}:^4A_2 \rightarrow {}^4T_2$, respectively, which are coincident with the characteristic absorption peaks of $Er^{3+}$, so that Er,Cr:YSGG can generate a 2780 nm CW laser output; therefore, the Er,Cr: YSGG laser is also known as a "water laser", which is mainly used in dental surgery [16]. Er,Yb: YSGG crystal has a high absorption capacity at the 980 nm LD pump source because of its high transfer efficiency between Nd and Er [17,18]. In contrast, the deactivation ions like $Pr^{3+}$, $Ho^{3+}$ and $Eu^{3+}$ can help to solve the self-terminating behavior and low laser efficiency which are caused by the lifetime of lower-level $^4I_{13/2}$, which is much longer than that of the upper level $^4I_{11/2}$. The main reason for this is that deactivation ions can help to reduce the 1.5 μm competitive luminescence and the difficulty of population inversion; the particle population in lower-level Er3+:4I13/2 is reduced by up-conversion due to the energy transfer between the two similar energy level, which makes the operation Er3+: 4I11/2 → 4I13/2 transition more possible and efficient [19–21]. According to the doping ion's energy level structure, $Nd^{3+}$ also can be utilized to transfer energy to $Er^{3+}$ effectively; thus, Er,Nd: YSGG will have a high absorption efficiency with an LD pump source. In addition, $Nd^{3+}$ can facilitate inducing population inversion to achieved mid-infrared emission. The growth of Er, Nd: YSGG single crystal fibers will benefit the optimization of the Er laser [22,23]. With the development of crystal growth methods, various functional crystal materials are grown to apply to different applications.

In this work, considering the volatilization of $Ga_2O_3$ during growth, we used the μ-PD to grow high-quality Er, Nd: YSGG SCFs for the first time. Then, the concentration distribution of $Er^{3+}$ was used to evaluate the crystal quality. And, the $Nd^{3+}$ was introduced into the Er: YSGG for studying the deactivating effect on 3.0 μm laser output. The relevant investigation of the fluorescence emission spectra and fluorescence lifetime were calculated simultaneously. Our works prove that Er, Nd: YSGG SCFs are potential materials for further 3 μm laser generations.

## 2. Methods

### 2.1. Crystal Fiber Growth

The polycrystalline materials are prepared by a conventional solid-state reaction, and an excess of 2 wt.% $Ga_2O_3$ is added to the initial raw materials to maintain the stoichiometric proportion due to the loss of volatilization during growth [24]. In our experiment, high-quality Er: YSGG crystal is selected as the seed crystal, and a home-made μ-PD furnace with inductive RF heating and chamber and a Φ3 mm die Ir crucible is used. The crucible can determine the diameter of SCF and affect the liquid/solid interface, while gas such as Argon should be used as the growth atmosphere to protect Ir from oxidation accordingly. In order to form a suitable temperature gradient, $ZrO_2$ insulation, alumina and quartz are placed to keep stable liquid/solid interfaces during growth [25]. As mentioned above, the molten material can stay stable in the crucible under the combined actions of surface tension and gravity, when the seed crystal slowly touched the bottom of the crucible,

and the melted material begins to flow through the micro capillary and crystallize on the oriented seed crystal. After the solid–liquid section maintains a stable shape, the seeding process is completed and the single crystal fiber can begin to grow with the set program. As shown in Figure 1, we employed a CCD camera to observe the liquid–solid interface and grown fibers. We can adjust the growth parameters by observing the liquid/solid interfaces, the seed and the as-grown fiber by real-time imaging.

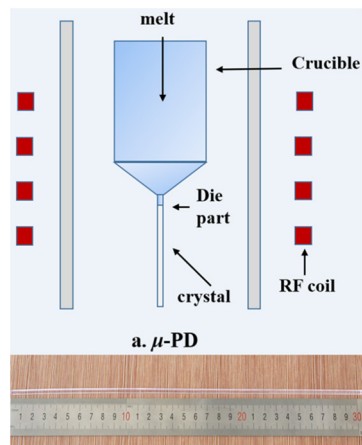

**Figure 1.** Schematic of μ-PD and as-grown crystal.

*2.2. Spectroscopic Properties*

Spectrophotometry (FLS920, Edinburgh Instrument, Edinburgh, UK) is performed to record the absorption spectra at room temperature, where the polished wafers with $\Phi$ 2 × 2 mm$^3$ are used as the samples. Before the polishing, we tested that the Mohs hardness of YSGG was 7.02, which is lower than 8–8.5 of YAG crystal, so the breakage at the end face was avoided through careful polishing and optimized polishing process. Moreover, the same samples are used to measure the fluorescence spectra and lifetime under a 970 nm LD by a spectrophotometer (FLS980, Edinburgh Instrument.UK). The surface of test samples is polished and the samples are all $\Phi$ 2 × 20 mm$^3$. The fitting fluorescence lifetime is calculated by fitting the test curve, and the fitting formula is $\tau_i = \frac{\sum \alpha_i \tau_i^2}{\sum \alpha_i \tau_i}$. ($\alpha_i$ representing the fractional weights of the various decay time components, and $\tau_i$ is the result of the exponential fitting).

### 3. Results and Discussion

*3.1. The Growth of Crystal Fibers*

In our experiment, the observation window was usually covered by the volatilization of $Ga_2O_3$ at the end of growth, so we utilized argon gas with 50% carbon dioxide to protect the crystals from corrosion by the volatilization of $Ga_2O_3$. The main principle was that $CO_2$ can produce $O_2$ at high temperatures, which can provide oxygen partial pressure to inhibit the volatilization of gallium oxide. In addition, CO produced by the reaction also can protect the iridium gold crucible from oxidation. Compared with a pure argon atmosphere, the argon gas with 50% carbon dioxide will make the observation windows cleaner at high temperatures during growth, when there are only few white volatiles deposited in the cold temperature field outside after the end of growth.

Additionally, the growth rate is an important factor in achieving high-quality SCFs, because the crystal grown at 1 mm/h or lower always represents a rough surface and is non-transparent. After the analysis of the component of as-grown crystal, it is mainly caused by the corrosion of iridium gold and gallium oxide, and the longer the crystal stays in the cold melting zone, the more serious the thermal corrosion will be. Moreover, we also observed that at a higher temperature near the crucible at the bottom of the crystal, the crystal has a higher quality, which is because the crystal near this section is closer to the hot

melting zone. In order to obtain a high-quality and transparent crystal, we changed the growth rate by up to 5 mm/h based on the existing temperature field gradient, because a high rate can help the grown crystals move away from the melting zone as far as possible to avoid thermal erosion.

With the optimized parameters, 30at.% Er, 2at.% Nd: YSGG and 30at.% Er, 5at.% Nd: YSGG with diameters of 2 mm were successfully obtained using the μ-PD method. The SCFs' lengths could reach to 100 mm, and the crystals were light purple due to the erbium and neodymium (Figure 2b,c). As contrasted with co-doped crystals, we also prepared 30at.% Er: YSGG with the same conditions, and the color was pink. These SCFs were apparently transparent after the optimization of the growth conditions.

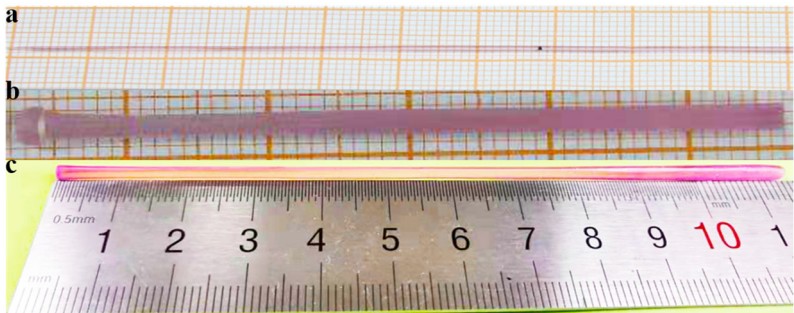

**Figure 2.** (**a**) 30at.% Er: YSGG; (**b**) 30at.%Er, 2at.% Nd: YSGG; (**c**) 30at.% Er, 5at.% Nd: YSGG.

### 3.2. Concentration Distribution

The Er: YSGG was well polished to measure the concentration distribution. The concentration distribution of $Er^{3+}$ and $Nd^{3+}$ along the radial direction were also measured using Electron Probe microanalysis (EPMA). It is worth mentioning that the measurement could only show a relative result because of the lack of a standard YSGG sample. The selected modes were a line scanning test and a mapping test. We collected the data at an interval of 0.5 s. As seen in Figure 3, the distribution of active ions was homogeneous along the cross section of the sample, and where $Er^{3+}$ or $Nd^{3+}$ ions occupied the position of $Y^{3+}$ ions. Hence, it proves that μ-PD is an effective method to grow high-doping laser crystals.

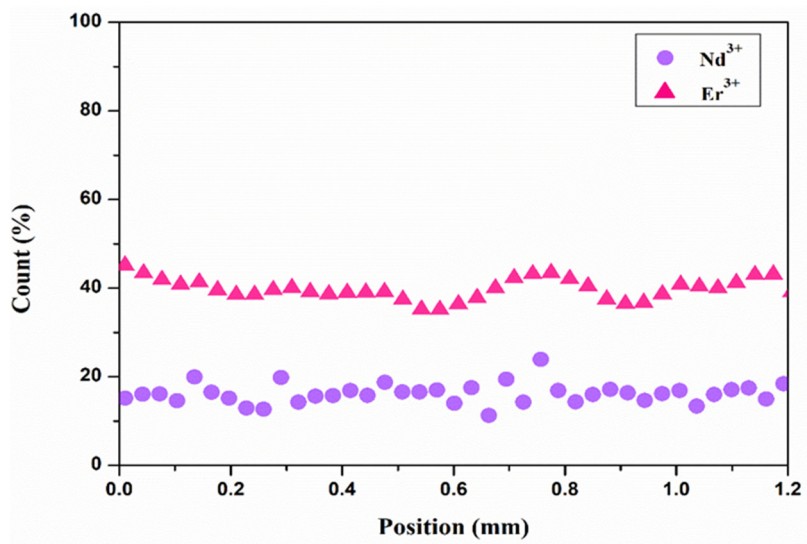

**Figure 3.** Concentration distribution of $Er^{3+}$ ions and $Nd^{3+}$ ions in 30at.% Er, 2at.% Nd: YSGG crystals.

### 3.3. Spectroscopic Properties

$Nd^{3+}$ had a similar energy level structure as $Er^{3+}$, which makes the energy transfer accessible and causes less lattice distortion in co-doped crystal. Therefore, it could be

revealed by the analysis of the spectroscopic properties of the $Er^{3+}$- and $Nd^{3+}$-doped YSGG. The room temperature (RT) absorption spectra of samples in the range of 300–1500 nm are shown in Figure 4a. The absorption peak at 967 nm matches well with the emission of the common InGaAs laser diode pumping source. Furthermore, the energy gap of Nd: $^4I_{9/2} \rightarrow {}^4F_{5/2}, {}^2H_{9/2}$ was close to the energy of Er: $^4I_{15/2} \rightarrow {}^4I_{11/2}$; the small energy gap makes energy transfer efficient in co-dopant crystals. It is can be seen in Figure 4b, c that the absorption peaks at 808 nm are much stronger compared with the absorption peak at 967 nm. Hence, we could use 808 nm LD as the pump source in future experiments. In conclusion, an appropriate $Nd^{3+}$ will be a remarkable ion to provide crystals with a high-absorption efficiency at the pump during the laser operation.

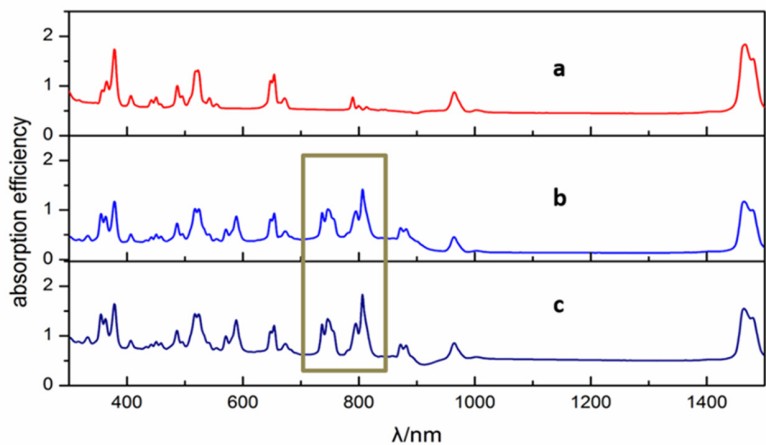

**Figure 4.** Absorption spectra of as-grown crystal in 300–1500 nm, (**a**) 30at.% Er: YSGG (**b**) 30at.% Er, 2at.% Nd: YSGG; (**c**) 30at.% Er, 5at.% Nd: YSGG.

In this experiment, the characterization of the fluorescence emission of high concentration erbium-doped crystals near 3 µm provides conditions for future laser experiments. We tested the emission spectra of Er:YSGG at room temperature using a laser pump source excited by 970 nm LD. The measurement band was around 3.0 µm, and the sample size was $\Phi\,3 \times 2\,mm^3$. It can be seen from Figure 5 that the main mid-infrared emission band centered at 2638 nm and 2817 nm. The results of the samples show that the emission band was centered around 2790 nm, which is consistent with the absorption peaks of the hydroxide ion. Meanwhile, the multispectral bands indicate that the crystal is potentially tunable to obtain a 3 µm laser output.

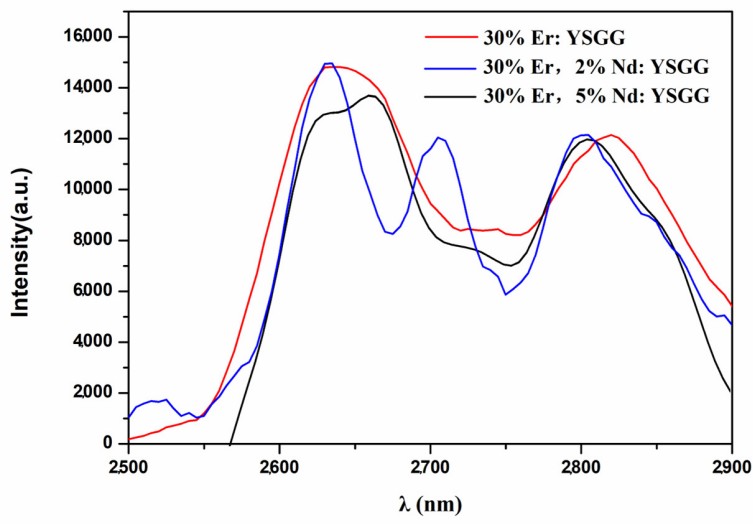

**Figure 5.** Emission spectra of crystal excited by 970 nm.

In order to explore the deactivating effect of $Nd^{3+}$, the detail of the fluorescence lifetime was analyzed. $Er^{3+}$-doped crystals were used to obtain laser outputs of around 3 μm and 1.5 μm, but the operation that obtains a 1.5 μm laser output is much easier to perform than that of a 3 μm laser output, because the lower-level lifetime of $^4I_{13/2}$ is too long, so the population inversion is hard to hold. So, we studies the influence of $Nd^{3+}$ on fluorescence lifetime to solve the problem. Compared with other crystals, as can been in Table 1, the relevant lifetimes of lower-level $^4I_{13/2}$ and the upper-level $^4I_{11/2}$ of Er: YSGG were calculated to be 6.05 ms and 1.6 ms, respectively, while the corresponding lifetime of Er: YAG was 7.25 ms and 0.1 ms. The decrease in the energy level difference between $^4I_{13/2}$ and $^4I_{11/2}$ makes Er: YSGG a potential material for achieving a 2.79 μm laser. In contrast, we studied the fluorescence decay curves of doped samples. Figure 6 shows the as-calculated lifetime and the 970 nm LD that we used as an excitation source. The calculated lifetimes are of sample b: $^4I_{13/2}$ (0.384 ms) and $^4I_{11/2}$ (0.194 ms), and sample c: $^4I_{13/2}$ (0.245 ms) and $^4I_{11/2}$ (0.176 ms). Our grown crystals had advantages of a smaller energy level difference compared with other crystals, which is beneficial for reducing the number of particles in $^4I_{13/2}$ to maintain population inversion [26].

We listed some research and compared the deactivating effect of introducing the Nd. As can be seen in Table 2, the Er/Nd co-doped crystals all have shorter lifetimes than Er:$^4I_{13/2}$; the difference in the lifetimes of two energy levels will clearly reduce. It will accelerate the relaxation of particles in the lower-level to make the transition of $Er^{3+}$: $^4I_{11/2} \rightarrow {}^4I_{13/2}$ more probable, which will be helpful in further laser experiments of around 3 μm.

**Table 1.** The fluorescence lifetime of several garnet crystals [25–30].

| | Er: $^4I_{13/2}$ Lifetime (ms) | Er: $^4I_{11/2}$ Lifetime (ms) | Difference (ms) | Methods |
|---|---|---|---|---|
| 30at.% Er: YAG ($Y_3Al_5O_{12}$) | 7.25 | 0.1 | 7.15 | CZ |
| 30at.% Er: GGG ($Gd_3Ga_5O_{12}$) | 4.86 | 0.9 | 3.96 | CZ |
| 20at.%,5at.%Er: LuSGG ($Lu_3Sc_2Ga_3O_{12}$) | quenched | 0.38 | / | CZ |
| 30at.% Er: SGGM ($SrGdGa_3O_7$) | 6.07 | 0.63 | 5.44 | CZ |
| 30at.% Er: YSGG | 6.05 | 1.6 | 4.45 | μ-PD |
| 30at.% Er: 2atNd: YSGG (this work) | 0.38 | 0.19 | 0.19 | μ-PD |
| 30at.% Er: 5atNd: YSGG (this work) | 0.26 | 0.18 | 0.08 | μ-PD |

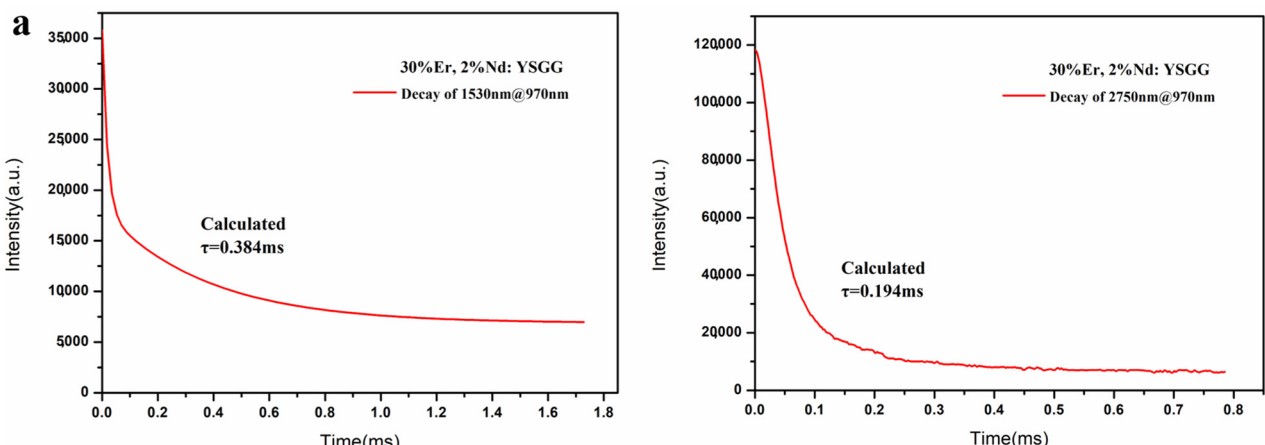

**Figure 6.** *Cont.*

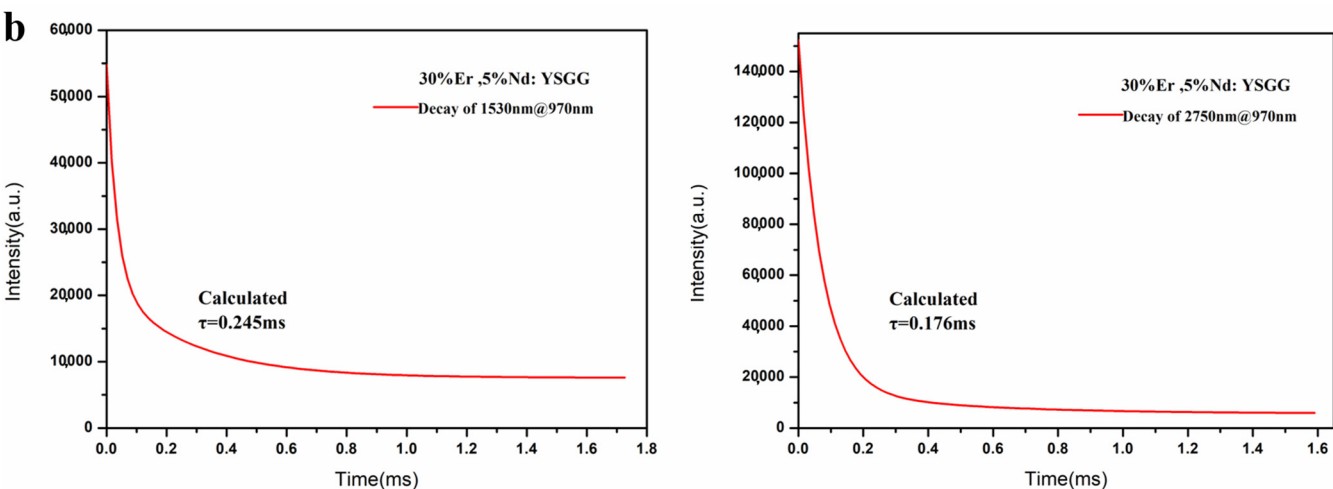

**Figure 6.** Fluorescence decay curves of Er: $^4I_{13/2}$ in 1530 nm and $^4I_{11/2}$ in 2790 nm excited by 970 nm, (**a**) 30at.% Er, 2at.% Nd: YSGG; (**b**) 30at.% Er, 5at.% Nd: YSGG;.

**Table 2.** The fluorescence lifetime of several Er/Nd-doped crystals.

| Crystals | Er: $^4I_{11/2}$ $\tau$ (ms) | Er: $^4I_{13/2}$ $\tau$ (ms) | Reference |
|---|---|---|---|
| Er: BLGO | 0.70 | 9.76 | [31] |
| Er/Nd: BLGO | 0.17 | 4.26 | [32] |
| Er: SLGO | 0.71 | 9.74 | [33] |
| Er/Nd: SLGO | 0.55 | / | |
| Er: PbF2 | 5.89 | 12.99 | [34] |
| Er/Nd: PbF2 | 5.26 | 2 | |
| Er: GYAP | 0.84 | 4.02 | [22] |
| Er/Nd: GYAP | 0.83 | 0.93 | |
| Er: YSGG | 1.6 | 6.05 | This work |
| Er/Nd: YSGG (2at% Nd) | 0.19 | 0.38 | |
| Er/Nd: YSGG (5at% Nd) | 0.18 | 0.26 | |

Furthermore, the possible energy transfer processes that could be explained for the mechanism are shown in Figure 7. Based on the energy level, there are such transfers as $Er^{3+}$: $^4I_{13/2} \rightarrow Nd^{3+}$: $4I_{15/2}$ and the energy transfer up-conversion process $Er^{3+}$: $^4I_{13/2}$ + $Nd^{3+}$: $^4I_{15/2} \rightarrow Er^{3+}$: $^4I_{15/2}$ + $Nd^{3+}$: $^4F_{5/2}$, $^2H_{9/2}$, respectively. The combined effect induces a relatively low number of particles in $^4I_{13/2}$, which enable it obtain a 2.79 μm fluorescence emission easily [27–30]. As a consequence, the data mentioned above, indicating the significant effect of Er–Nd-doping of the YSGG, can also provide a better understanding to explore a high 2.79 μm laser output.

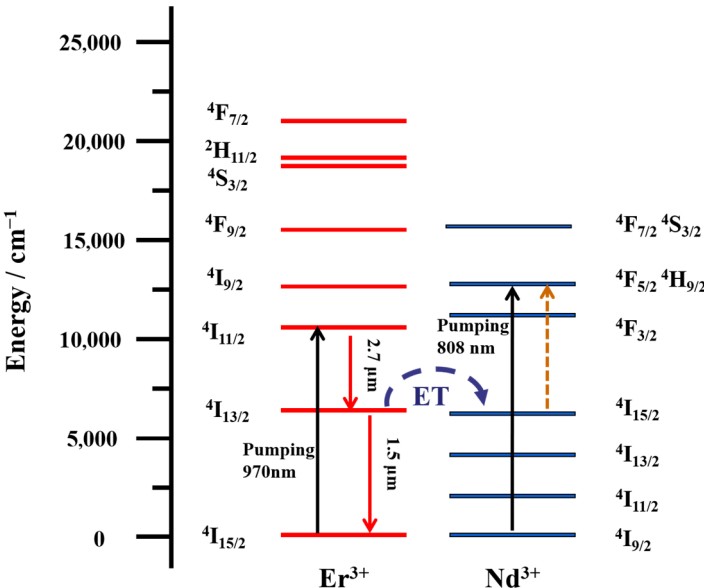

**Figure 7.** Energy-level and energy-transfer schematic of $Nd^{3+}$- and $Er^{3+}$-co-doped systems.

## 4. Conclusions

In this work, high-quality Er, Nd: YSGG SCFs were successfully grown using the μ-PD method with <111>-oriented YSSG seed crystals. The growth parameters were investigated systemically to achieve a high quality SCFs. In addition, $Nd^{3+}$ was designed and introduced into the matrix to optimize the optical properties of Er: YSGG. Furthermore, the lifetime of Er: $^4I_{13/2}$ was found to be 0.384 ms and 0.245 ms in co-doped crystal. Compared with Er: YSGG, its lower-level lifetime was decreased sharply due to the introduction of $Nd^{3+}$, which can be attributed to the energy transfer process and the energy transfer up-conversion process between $Nd^{3+}$ and $Er^{3+}$. All of these results indicate that Er: YSGG SCFs have great potential applications in the field of high-power lasers of around 3 μm.

**Author Contributions:** Crystal growth and fluorescence decay curves, B.W.; measurement of concentration distribution, B.W. and T.W.; measurement of absorption spectra and emission spectra, J.Z., N.J. and M.W.; writing, B.W.; funding acquisition, Z.J. and Z.W. All authors have read and agreed to the published version of the manuscript.

**Funding:** This work was supported by National Natural Science Foundation of China (NSFC) (11974427, 12004431), Science and Technology Innovation Program of Hunan Province (2021RC4027), State Key Laboratory of Pulsed Power Laser Technology (SKL-2021-ZR01), National Natural Science Foundation of China (52202008), and Natural Science Foundation of Shandong Province (ZR2022QE013).

**Data Availability Statement:** Data are contained within the article.

**Conflicts of Interest:** The authors declare no conflict of interest.

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
