# Peer review of "The Growth and Spectroscopic Properties of Er, Nd: YSGG Single Crystal Fibers"

_crystals, doi:10.3390/cryst13121646_

Round 1
Reviewer 1 Report
Comments and Suggestions for Authors
Wu et al. presented short manuscript concerning growth and spectroscopic characterization of Er, Nd: YSGG single crystal fibers. The Introduction section provides sufficient reasoning for the reported studies. Although, in my opinion, the manuscript needs a more detailed description of the growth, composition and structure of the crystals. Below are some suggestions.
1. What was the temperature regime during crystal growth? It would be useful to describe briefly the growth procedure.
2. Did the authors estimate the distribution coefficients of Er and Nd impurities?
3. The distribution of impurity ions was only measured in the radial cross section, wasn't it? What was the distribution along the crystal?
4. Was the composition of the growth crystals stoichiometric?
5. Did the authors perform X-ray phase analysis?
6. A brief comparison of the data obtained by the authors with the state-of-the-art literature would be useful.
Comments on the Quality of English LanguageModerate editing of English language required.
Reviewer 2 Report
Comments and Suggestions for Authors
Page 3 line 112: please introduce the equation and its terms.
Please ensure that you have introduced all acronyms upon first use.
You flip back and forth between nm and cm-1 when describing the energy levels and emission lines. Would it be better to use one or the other for consistency? Please consider revising.
Page 6: Why is it important to "deactivate" the emission line lifetimes? What is gained by this and what would happen if his did not occur? Please clarify.
Line 196: When you say "particles" do you mean excited states? Please consider revising.
Table 1: please ensure consistency and necessity of the significant figures displayed for all of the values of lifetimes.
Figure 6 a and b: These plots all look similar, would it be helpful to have them all with the same y-axis magnitudes so it would be easier to distinguish between them? Please clarify their significance to the work.
Comments on the Quality of English Languageminor revisions to the English language would be beneficial to the reader
Round 2
Reviewer 1 Report
Comments and Suggestions for Authors
The authors have commented on all suggestions. The manuscript can now be published in the journal.
Comments on the Quality of English LanguageModerate editing of English language required